# Expression Level of Mature miR172 in Wild Type and *StSUT4*-Silenced Plants of *Solanum tuberosum* Is Sucrose-Dependent

**DOI:** 10.3390/ijms22031455

**Published:** 2021-02-01

**Authors:** Varsha Garg, Aleksandra Hackel, Christina Kühn

**Affiliations:** Department of Plant Physiology, Institute of Biology, Humboldt-Universität zu Berlin, Philippstr. 13, Building 12, 10115 Berlin, Germany; varsha.garg.1@hu-berlin.de (V.G.); hackelal@hu-berlin.de (A.H.)

**Keywords:** photoperiodic control, flowering, tuberization, miR172, sucrose transporter, sucrose signaling

## Abstract

In potato plants, the phloem-mobile miR172 is involved in the sugar-dependent transmission of flower and tuber inducing signal transduction pathways and a clear link between solute transport and the induction of flowering and tuberization was demonstrated. The sucrose transporter StSUT4 seems to play an important role in the photoperiod-dependent triggering of both developmental processes, flowering and tuberization, and the phenotype of *StSUT4*-inhibited potato plants is reminiscent to miR172 overexpressing plants. The first aim of this study was the determination of the level of miR172 in sink and source leaves of *StSUT4*-silenced as well as *StSUT4*-overexpressing plants in comparison to *Solanum tuberosum* ssp. Andigena wild type plants. The second aim was to investigate the effect of sugars on the level of miRNA172 in whole cut leaves, as well as in whole in vitro plantlets that were supplemented with exogenous sugars. Experiments clearly show a sucrose-dependent induction of the level of mature miR172 in short time as well as long time experiments. A sucrose-dependent accumulation of miR172 was also measured in mature leaves of *StSUT4*-silenced plants where sucrose export is delayed and sucrose accumulates at the end of the light period.

## 1. Introduction

Sugar accumulation and starch biosynthesis in mature leaves follow day time dependent oscillations. Whereas the accumulation of transitory starch in leaves is strongly light-dark-dependent since the rate-limiting enzyme in starch biosynthesis, the ADP-glucose pyrophosphorylase (AGPase) is regulated by light, the accumulation of soluble sugars in mature leaves oscillates during the day with maximum levels in the middle of the light period [1]. In accordance with this sugar accumulation the transcript amount of sucrose transporters follow a circadian rhythm and continue oscillation even under constant light conditions [1]. Whereas StSUT1, the main phloem loader in potato, show maximum levels 6 h after light set on (and circadian evening elements, EE) were found in the promoter region of this gene according to Harmer and Kay [2]. In turn, the *StSUT2* gene show maximum levels in the early morning (and circadian morning elements (ME) were found in the cis-regulatory upstream region of the gene).

Sucrose transporter StSUT4 seems to be involved in sugar-dependent signaling processes and *StSUT4*-inhibited potato plants revealed a pleiotropic phenotype regarding carbon translocation and accumulation. Potato plants with down-regulated *StSUT4*-expression are early flowering and develop tubers even under non-inductive long day (LD) conditions [1]. They show reduced internode elongation and produced reduced amounts of ethylene [3]. *StSUT4*-RNAi plants are far-red insensitive and do not show shade avoidance response if shaded by neighboring plants or if grown under far red enrichment. Sucrose efflux from mature leaves is significantly increased at the end of the light period and increased export of photo-assimilates from leaves is accompanied by increased amount of soluble sugars or starch in sinks such as developing tubers [1,3]. Increased sucrose export is not necessarily correlated with a reduced amount of sugars in the leaves. Since the shade avoidance syndrome (SAS) is under control of phytochrome B (phyB) [4], the sucrose transporter *StSUT4* is expected to be one of the phyB downstream signaling targets of this photoreceptor, and the expression of all three sucrose transporter genes of potato plants seem to be under control of the circadian clock [1].

In turn, in Arabidopsis, the involvement of miR156 and miR172 and their appropriate target genes in the induction of flowering under long day (LD) conditions is well investigated. Recent advances showed that the level of miR172 and miR156 which both are involved in regulating the transitions between developmental stages, i.e., from the juvenile to adult stage as well as in the transition from the vegetative to the generative phase (for a review, see [5]) seem to be sugar dependent in feeding experiments [6,7]. It has been shown that miR156 is repressed by glucose, as well as by sucrose. It is also shown in Arabidopsis that the age-dependent miR156 integrates sugar-dependent information about the nutritional status of the plant via the signaling molecule trehalose-6-phosphate with the age pathway for flower induction [8].

Only little is known about the role of miR172 and miR156 in Solanaceous plants. One flower inducing pathway via the florigenic protein flowering locus T (in the following called FT protein) involves the two microRNAs 156 and 172 and expression of FT is repressed by two AP2 transcription factors (SNZ and SMZ) which represent targets of miR172. Here, the induction of the florigenic FT gene occurs in a CONSTANS independent manner and seems to be mainly triggered via the sugar status in leaves. In potato, flowering and tuberization are triggered by the florigen and the tuberigen and both show similarities to flowering locus T from Arabidopsis. Potato plants possess as many as nine different FT homologues which are expressed in a photoperiod-dependent manner [9].

The aim of this study is the quantification of miR172 in the leaves of *StSUT4*-silenced and leaves of *StSUT4*-overexpressing plants in comparison to wild type plants. The question is whether sugar levels have an impact on the level of mature miR172 both in whole cut leaves as well as in whole in vitro plantlets that were supplemented with exogenous sugars. Our intention was to investigate whether similar mechanisms of sugar repression of miRNAs are valid for potato plants as previously observed in Arabidopsis.

## 2. Results

### 2.1. StSUT4-RNAi Plants Are miR172 Over-Expressors

We were interested in quantification of the level of miRNA172 in StSUT4-silenced plants since morphological observations revealed that the phenotype of *StSUT4* silenced plants is very similar to what was observed for miR172 overexpressing plants [10] regarding internode elongation, early flowering, tuberization under non-inductive conditions, etc., [1]. Comparing source leaf RNA, we observed a significantly increased level of miR172 in one out of two independent transformant lines with reduced levels of *StSUT4* expression (RNAi 2/5 and RNAi 2/16) compared to WT plants (Figure 1A,B). As a positive control, 35S::miR172 overexpressing potato plants have been used that show an even stronger upregulation of miR172 accumulation (Figure 1A). No significant increase in miR172 levels was observed in StSUT4-GFP overexpressing plants (Figure 1B).

Interestingly, the level of miR172 was significantly up-regulated in two independent sets of *StSUT4* silenced plants: both *StSUT4*-RNAi plants from *Solanum tuberosum* ssp. tuberosum Désirée (Appendix A) as well as from *Solanum tuberosum* ssp. Andigena (Figure 1A,B), which is strongly photoperiod-dependent with respect to tuberization, showed significant up-regulation of miR172 accumulation in source leaves (Figure 1A,B). In sink leaves however, the level of miR172 is not increased in *StSUT4*-RNAi plants (RNAi 2/5 and RNAi 2/16), but in *StSUT4*-GFP overexpressing plants (StSUT4-GFP, Figure 1C). Further analysis of the level of miR172 in the shoot apical meristem of *StSUT4*-RNAi plants revealed increased levels of miRNA sink organs as well (Appendix A).

Detailed analysis of miR172 levels during the light period in potato wild type and *StSUT4*-silenced plants indicated a diurnal oscillation of this miRNA during the light period (Appendix A) when soluble sugar levels are increasing as well [1].

This might explain the similar phenotypic changes of miR172 over-expressors and *StSUT4*-RNAi plants [1,10] and the question arose whether or not altered levels of soluble sugars in the leaves of *StSUT4* silenced plants are responsible for this increase in the level of mature miR172 in those plants.

### 2.2. miR172 Is Induced by High Levels of Sucrose Both in Long Time and Short Time Experiments

In order to test, whether the level of miR172 is affected by external sugar supply, we performed two different kinds of feeding experiment: one short time experiments using whole cut leaves from 6 weeks old adult plants grown under greenhouse conditions (Figure 2), or alternatively whole plantlets grown under sterile in vitro conditions in the phytochamber for 21 days on medium supplemented with different carbon sources (Figure 3).

The slight increase in miR172 transcript amounts during the short time experiment shown in Figure 2 when sugars are omitted from the medium or sorbitol is supplied might be explained by diurnal oscillation during the day. In the presence of 100 mM of sucrose a significant increase in the level of miR172 is detectable after 2 and 3 h of incubation (Figure 2) suggesting that miR172 expression is sucrose-inducible. Glucose, as well, showed a transient increase in miR172 after 3 h of incubation (Appendix A).

Additionally, in plants cultured in vitro for longer time (Figure 3), the amount of miR172 is significantly higher if plants were grown on sucrose containing medium. In both cases, the amount of miR172 is more than doubled compared to the control treatment without any sugar. Whereas glucose application showed a transient increase in miR172 after 3 h of incubation in the short time experiment (Appendix A), no significant increase in the level of miR172 was observed in the long time experiment (Figure 3).

### 2.3. In Vitro Plants Grow Optimal in the Presence of Sucrose

In order to test whether plant’s performance depend on the carbon source, we propagate potato cuttings under sterile in vitro conditions on different sugars or without any additional sugar supply in the medium. After 21 days of growth on Murashige and Skoog (MS) Medium, the plantlets growing on 80 mM sucrose containing medium behaved optimal with maximum plant height and maximum root length (Figure 3A,B), whereas plants grown on sorbitol containing medium or without any sugar supply remained much smaller.

## 3. Discussion

Potato plants overexpressing miR172 show graft-transmissible effects on flowering and tuberization, including promotion of flowering, acceleration of tuberization under moderately inductive photoperiods, induction of tuber formation under long days and shorter internode elongation [10]. Very similar phenomena were observed in *StSUT4*-silenced potato plants [1]. Therefore, we were interested in the level of this miRNA is the transgenic *StSUT4*-silenced, as well as in *StSUT4*-overexpressing potato plants. Our results demonstrate increased levels of miR172 in *StSUT4*-silenced source leaves, whereas in sink leaves this miR172 increases when *StSUT4* is overexpressed (Figure 1). This argues for StSUT4 acting upstream of miR172 in a sink-source dependent manner.

In sugar feeding experiments, an induction of miR172 in response to sucrose supply was observed in two kinds of experiments: direct sucrose feeding to cut petioles of adult leaves, as well as in a long term feeding experiments over a period of three weeks with in vitro grown plants (Figure 2 and Figure 3). In both experiments, the miR172 expression was induced by sucrose application (Figure 2 and Figure 3).

At the end of the light period, the efflux of sucrose was strongly enhanced in *StSUT4*-inhibited potato plants [1]. In parallel, the level of soluble sugars that is decreased during the day, is also increased to high levels at the end of the light period (15 h Zeitgeber time, [1]). The miR172 quantification shown in Figure 1 would be consistent with increased levels of soluble sugars in source leaves as we measured at the end of the light period. The levels of soluble sugars and starch in sink leaves of StSUT4-silenced plants is rather increased [1] and sucrose peaks in the shoot apical meristem is shifted to earlier time points in the transgenic plants [1] which might be the reason for early flowering, and perhaps also for an earlier onset of tuberization in these plants.

At least for the soybean miR172, it is known that expression follows a diurnal rhythm [11] and it cannot be excluded that miR172 in potato oscillated as well diurnally. The slight increase in miR172 transcripts during the short time experiment shown in Figure 2 in samples without sugar or treated with sorbitol argues for such a diurnal change of the level of miR172 during the day (Appendix A). Sucrose plays an important role in the entrainment of the circadian clock [12]. It cannot be excluded that changes in the level of miR172 are due to a shift in oscillation since the diurnal changes of soluble sugars is also shifted in StSUT4-silenced plants arguing for a disturbed entrainment of the circadian clock [1].

miR172 and miR156 are often engaged in identical regulatory pathways changing in opposite directions during plant development. In tomatoes, the overexpression of sly-miR156a partially phenocopies the phenotype of *sft* mutants, with respect to late flowering, more abundant leaves, dwarfism, bushier structure, etc. The SINGLE FLOWER TRUSS (SFT) transcript amount is significantly down-regulated in 35S::miR156a overexpressing tomato plants suggesting that SFT is among the targets of miR156a. Comparing the effects of miR156 overexpression in Arabidopsis, rice, maize, and tomatoes, an evolutionary conserved function for miR156 was postulated [13].

In potato plants, miR156a was shown to be expressed mainly under long day (LD) conditions in leaves, whereas expression in stolons is higher under short day (SD) conditions [14]. miR156a is graft-transmissible and promoter analysis suggests a light-dependent expression [14]. In potato, miR156 seem to affect multiple growth traits as miR156 overexpression results in altered leaf and trichome morphology, aerial tuber production under SDs, delayed flowering, reduced stomatal density, reduced root biomass etc. It was excluded that miR156 acts as a repressor of tuberization and that miR156 function differs in LDs and in SDs [14]. In Arabidopsis, the level of miR156a and miR156c is negatively affected by sugar supply [6,7,15]. Further experiments are needed to answer the interplay between these two miRNAs in Solanaceous plants.

Regarding sucrose transporter StSUT4, it is assumed that StSUT4 acts as a pace-maker of circadian expression of sucrose transporters and sucrose efflux from leaves in a phyB-dependent manner. Via time-dependent effects on the sugar level in source and sink tissues, StSUT4 thereby affects the level of the phloem-mobile miR172 and further downstream signaling events, potentially involving targeting of AP2-like transcription factors.

## 4. Materials and Methods

### 4.1. Plant Material and Growth Conditions

*Solanum tuberosum* L. subspecies *andigena* line 7540 was used as wild-type potato. Plants were propagated from single-node stem cuttings on Murashige and Skoog (MS) medium containing either 16 g·L^−1^ glucose, sucrose, sorbitol, or no sugar at all and 2 g·L^−1^ agar (Duchefa Biochemie, Haarlem, The Netherlands) at 23 °C under LD conditions (16 h light and 8 h darkness). 35S::miR172 overexpressing plants were kindly provided by Paula Suarez-Lopez [10]. In order to analyze the level of mature miR172 in StSUT4-RNAi plants were designed stem loop primers for qPCR according to [9]. As a positive control was used miR172 overexpressing potato plants that overexpress miR172 under control of the constitutive CaMV35S promoter [9]. The experiments were conducted on the transgenic plants in order quantification of the level of miRNAs. Plant material (source and sink leaves) for analyses were collected from 6 weeks old plants grown under long day (LD) conditions in the green house. The following experimental variants were used: wild type (WT), StSUT4-RNAi plants (RNAi2/5; RNAi2/16), CaMV35S::miR172-overexpressing plants (miR172OE8; miR172OE22), and StSUT4-overexpressing plants (SUT4-GFP23, SUT4-GFP14). Yeast complementation experiments were performed in order to test whether C-terminal fusion of GFP to sucrose transporters affect sucrose transport capacity [16]. Sugar feeding experiments were performed with whole plantlets in long time experiments (3 weeks of growth with or without sugar) and whole cut leaves from 6 weeks old wild type potato plants in short time experiments (0 to 5 h incubation with or without sugar).

### 4.2. Plant Material for Sugar Feeding Experiments

Leaves of potato wild type plants grown under greenhouse conditions with a 16h light–8 h dark cycle were harvested from 6 week old plants. Petioles of detached leaves were re-cut while submerged in water, 2.5 mM EDTA was added to inhibit callose formation, and the cut petioles were transferred to the appropriate sugars solution containing 2.5 mM EDTA in addition. Sugar concentration was 100 mM glucose, sucrose, or sorbitol. The transcript amount of each miRNA was determined before the experiment (t0) and samples were collected at the indicated period of time (30 min, 1 h, 2 h, 3 h, and 4 h). Experiments were conducted from 10 am till 2 pm under greenhouse conditions. Quantification by qPCR was conducted as indicated below.

### 4.3. Analysis of miRNA Levels

RNA was isolated from leaves or whole sterile plants. RNA extraction was performed using TRIzol (Invitrogen, Carlsbad, California, United States) according to the manufacturer’s protocol. Reverse transcription was performed with the SuperScript Reverse Transcriptase (Thermo Fischer, Waltham, MA, USA) according to the manual. Reverse transcription was performed using oligo(dT) primers on approximately 1 µg of total RNA after digestion with RNase-free DNase I (Fermentas/Thermo Scientific).

For the quantification of small RNAs, total RNA was reverse-transcribed using SuperScript Reverse Transcriptase (Thermo Fischer) following the manufacturer’s protocol with 1 μL of a 1:1 mixture of oligo-(dT)_18_ and the miRNA-specific stem-loop RT primers used in the priming step. The stem-loop RT primer and the oligonucleotide pair used for qRT-PCR analysis are shown in Table 1.

Aliquots of 20 ng of cDNA and primer (Table 1) concentration of 250 nM were used for the subsequent qRT-PCR reaction. qRT-PCR was performed using ChamQ Universal 2× SYBR^®^Green Mix (Vazyme Biotech, Nanjing, China) reaction mixture in a Bio-Rad CFX System (Bio-Rad Laboratories, Feldkirchen, Germany) in 10 µL reaction volumes. Experiments were performed in biological triplicates.

The qRT-PCR reactions were performed in the following cycling conditions: activation for 10 min at 95 °C, denaturation at 95 °C for 15 s, annealing for 15 s at 60 °C and elongation for 15 sec at 72 °C, in a program of 45 cycles. Relative quantification of transcript amounts was always calculated in relation to the respective transcript level of appropriate reference genes (5S rRNA) and given as relative expression (2^(−ΔΔCt)^).

For normalization, 5S ribosomal RNA was reverse transcribed by stem-loop primer 5S rRNA stem loop primer and amplified by 5SrRNAfw and universal rev in the same conditions as for miR172 and miR156. PCR specificity was checked by melting curve analysis, and data were analyzed using the 2^−ΔΔCt^ method [17].

### 4.4. Primers for qPCR Analysis

### 4.5. Statistical Analysis

Statistical evaluation of real time data was performed with n between 6 and 12 individuals (three biological replicates and two technical replicates). To test whether phenotypical traits differ between treatments or whether the individuals of single transgenic lines differ significantly in their expression level from WT plants, factorial analysis of variance (ANOVA) was carried out in accordance with the experimental design with α= 0.05 using the SigmaPlot or Excel software. Significant differences are indicated by different letters and the SE is given.

## 5. Conclusions

The results of our study demonstrate considerably higher level of miR172 in several transformant lines with reduced levels of *StSUT4* expression compared to wild type plants. Moreover, sucrose had a positive impact on the accumulation of miR172 both in leaves of whole plantlets in long time experiments as well as in whole cut leaves in short time experiments. In fact, potato and Arabidopsis share similar signal transduction components regarding the flower inducing signaling pathway and miRNAs seem to play an important role in both pathways [18,19]. Sugar dependent effects on the expression level of the age-dependent miR156 are already described in case of Arabidopsis which with respect to flowering control is assumed to represent a long day plant, whereas potato with respect to tuberization is short day-dependent. Here, the level of miR172 which targets members of the AP2 transcription factors seems to be sugar-inducible as shown in short time and long-time experiments. The sugar status in leaves of potato, which is an important crop, has severe impact on downstream signaling events affecting flowering and tuberization. It is obvious that miR172 as well as sucrose transporter StSUT4 are involved in those pathways.

## Figures and Tables

**Figure 1 ijms-22-01455-f001:**
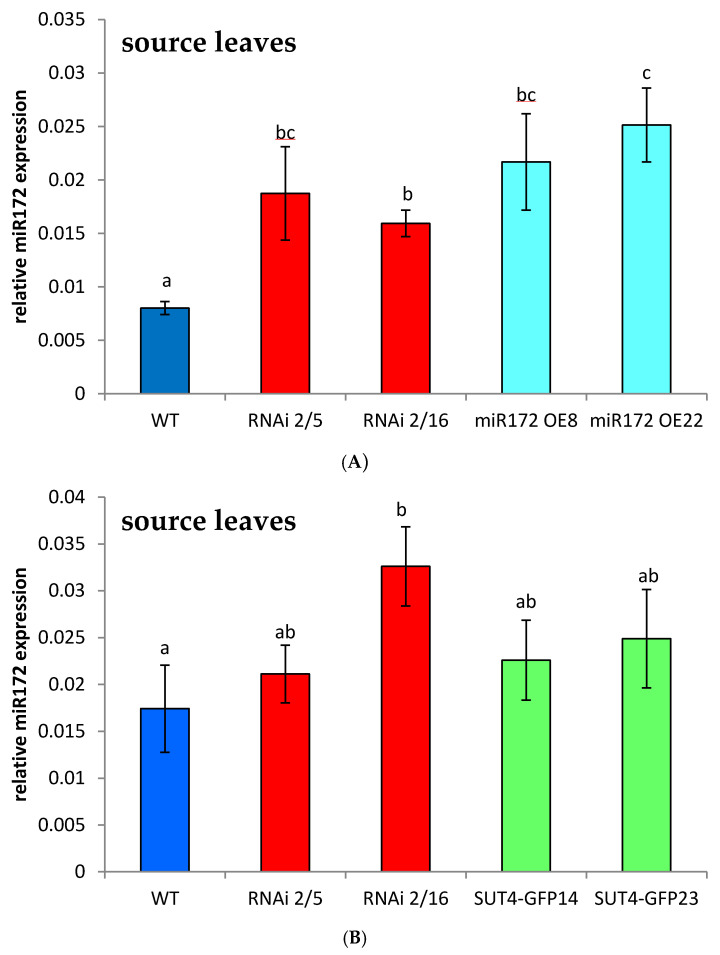
(**A**) Level of miR172 in source leaves of 35S::miR172 overexpressing potato plants (cyan bars) and *StSUT4*-RNAi plants (red bars) compared to *Solanum tuberosum* ssp. Andigena wild type plants (blue bar). 35S::miR172 overexpressing potato plants were previously shown to express increased amounts of miR172 [10] and were used as positive control. (**B**) Quantification of miR172 in source leaves of StSUT4-RNAi plants (red bars) and CaMV35S::StSUT4-GFP overexpressing plants (green bars). (**C**) Quantification of miR172 in sink leaves of StSUT4-RNAi plants (red bars) and CaMV35S::StSUT4-GFP overexpressing plants (green bars). Relative quantification was performed using 5SrRNA as a reference. StSUT4-RNAi plants were already described previously [1]. One way ANOVA was performed with (α = 0.05). Significant differences are indicated by different letters.

**Figure 2 ijms-22-01455-f002:**
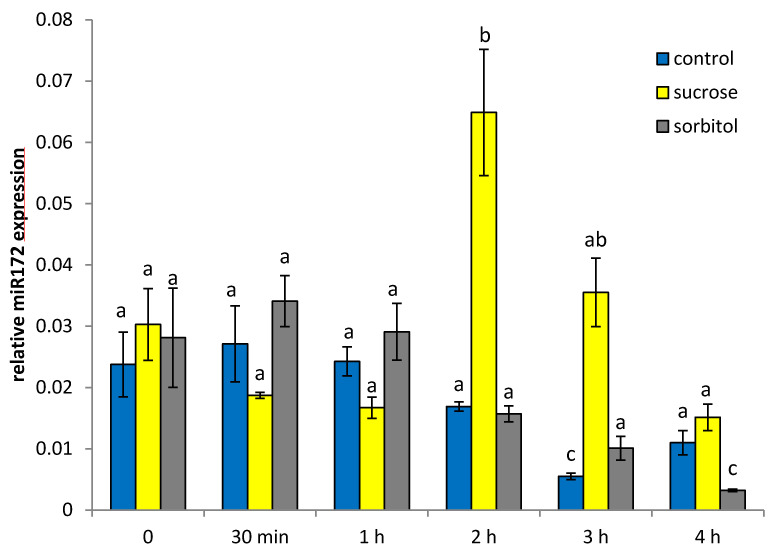
Quantification of miR172 in mature leaves depending on sugar supply in a short time experiment. Short time petiole experiment in the presence or absence of sugars was conducted with source leaves of 6 weeks old potato wild type plants (*Solanum tuberosum* variety Désirée). Sugars such as sucrose or sorbitol were supplied at a concentration of 100 mM each in 2.5 mM EDTA. Note that miR172 levels in the absence of sugars might oscillate diurnally during the day. Quantification was performed using 5SrRNA as a reference (α = 0.05). Significant differences are indicated by different letters and SE is given.

**Figure 3 ijms-22-01455-f003:**
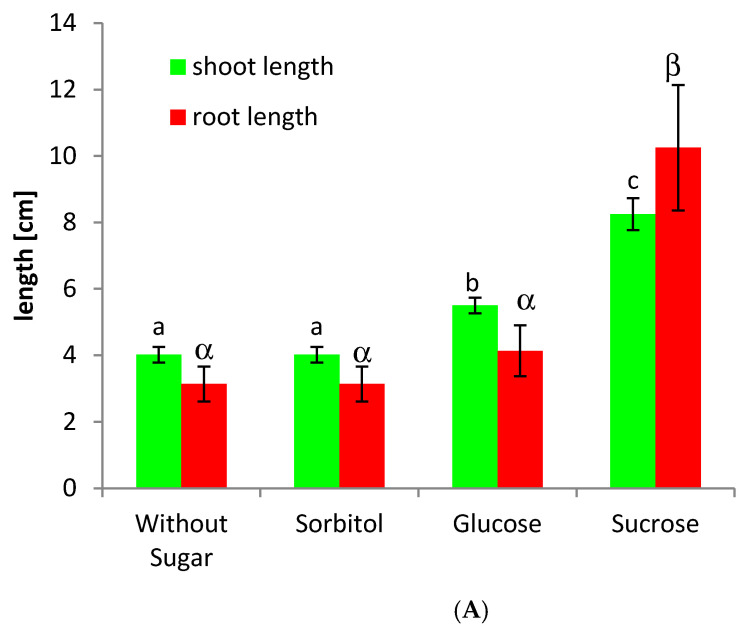
Growth and root morphology of potato wild type plants grown in vitro for 21 days on different carbon sources. Shoot and root length of six individual wild type potato plantlets were measured (**A**), as well as the total plant size (**B**) depending on the carbon source. Optimal growth and highest biomass production was observed when plantlets were grown on Murashige & Skoog (MS) medium supplied with 80 mM of sucrose. The standard error of the mean is given in (**A**) with *n* = 6 individual wild type plants. (**C**) Quantification of miR172 in a long time experiments by qPCR using whole plantlets, which photos are shown in (**A**) after 21 days of growth under sterile conditions. Whereas the effect of glucose and sorbitol on the transcript amount of miR172 is statistically not significant, the accumulation of miR172 transcripts is significantly higher when sucrose has been added in a concentration of 80 mM. Quantification of miRNAs was performed with *Solanum tuberosum* ssp. tuberosum plants using 5SrRNA as a reference (with *p* < 0.001). Significant differences are indicated by different letters.

**Table 1 ijms-22-01455-t001:** Primers used for quantification of miRNAs.

miR156 stem loop primer	GTCGTATCCAGTGCAGGGTCCGAGGTATTCGCACTGGATACGACGTGCTC
miR156 fw primer	GCGGCGGTGACAGAAGAGAGT
miR172 stem loop primer	GTCGTATCCAGTGCAGGGTCCGAGGTATTCGCACTGGATACGACAGGGAT
miR172 fw primer	CGGCGGTAGAATCTTGATGATG
5SrRNA stem loop primer	GTCGTATCCAGTGCAGGGTCCGAGGTATTCGCACTGGATACGACAGGGAT
5SrRNA fw primer	GGATGCGATCATACCAGCACT
Universal rev primer	GTGCAGGGTCCGAGGT

## Data Availability

The data that support the findings of this study are available from the corresponding author upon reasonable request.

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
