# Peer review of "Expression Level of Mature miR172 in Wild Type and *StSUT4*-Silenced Plants of *Solanum tuberosum* Is Sucrose-Dependent"

_ijms, 2021, doi:10.3390/ijms22031455_

Round 1
Reviewer 1 Report
This new version still suffers from several flaws that in my opinion make it not to be suitable for the publication in IJMS.
The main message of this paper is the StSUT4-mutant potato plants and miRNA172-mutant plants share the same morphological changes leading the authors to assume that StSUT4 might play a key role in sugar-mediated the regulation of miRNA172.
Unfortunately, the results depicted in the figure 1 do not definitely and undoubtedly support this hypothesis for many reasons. Please see below :
1) Page 88, there is mentioned that a striking increased level of miRNA72 in several independent transformant lines of StSUT4-silenced plants, while the results of the figure 1A indicate such an upregulation only for RNAi 2/5 line. No statistically significate augmentation was recorded with RNAi2/16 line of StSUT4 mutant.
2) Line 91, regarding source leaves, the authors had only tested one StSUT4-GFP overexpressing plants. By contrast, for sink organs, they studied two StSUT1-GFP ones. I strongly recommend that the authors merge the figure 1A and 1B and test, as it is the case for sink leaves, the tow StSUT4-GFP overexpressing plants (SUT4-GFP23 and SUT4-GFP14) in terms of miRNA172 expression.
3)Line 94, In order to strengthen the hypothesis that the expression of miRNA172 would be upregulated in StSUT4-inhibited plants, data from Solanum tuberosum ssp. tuberosum Désirée should be added either in the main manuscript or as supplemental data.
4) Line 97, the authors mentioned that the sink leaves exhibited the opposite pattern in terms of miRNA172 expression, because miRNA expression is decreased in StsSUT4-RNAi plants. When looking at the figure 1C, the expression of miRNA172 is almost similar to WT and no statistically significant difference is reported. In StSUT4-GFP overexpressing plants, the miRNA172 level is only higher for the line SUT4-GFP23. I am wondering why the authors did not monitor the expression of miRNA172 in young leaves of miRNA172 overexpressing plants, as positive control.
5) I did not understand what it expected from the data of the figure 4 in terms of sugar, StSUT4 and miRNA172 expression. Everyone knows that in vitro plants need some kind of sugars, and sugars affect their growth performance depending on many criteria including sugar concentration , the species. Please, could you explain the objective of this experiment.
6) Line 139, the authors stated that miRNA172 is sucrose-inducible. To support this, the effect of glucose on miRNA 172 expression is required, as they did for whole plantlets.
Author Response
Point-by-point Response to Reviewer #1
- The reviewer is right and we corrected the text line 88 accordingly.
- Line 91, the reviewer is right and we provide now a new figure 1B where the results for the source leaves of both StSUT4-GFP overexpressing lines , SUT4-GFP14 and SUT4-GFP 23 are shown as it was done for the level of miR172 in sink leaves.
- As suggested by the reviewer, we added a new supplementary figure with real time quantification of miR172 in StSUT4-silenced Solanum tuberosum Désirée plants in order to strengthen our hypothesis.
- Since the level of miR172 in sink and source organs of mir172 overexpressing plants are already analyzed in detail by Martin et al (2009), we focused on the expression level in sink and source organs in StSUT4 silenced Désirée and andigena potato plants and added new real time data, also from the shoot apical meristem (Supplements). We are now discussing that changes of the level of miR172 are most likely due to a shift in diurnal oscillation. The text in Line 97 was changed accordingly.
- Figure 4 is just the documentation of the in vitro grown plants used for the experiment shown in Figure 3. It is interesting to note, that growth performance is optimal in the presence of sucrose and we mentioned this growth behavior and biomass production in order to complete the experimental conditions.
- We now added data of glucose application in a short time experiment as supplementary Figure 3 showing a transient increase in miR172 level which is not seen in the long time experiments after 3 weeks of growth. This is now discussed.
Reviewer 2 Report
In the revised version of manuscript entitled "Expression level of mature miR172 in wild type and StSUT4-silenced plants of Solanum tuberosum is sucrose-dependent" intended for publication in International Journal of Molecular Sciences the Authors provide the most of required information and improvements. I think that manuscript still needs some improvements before publication.
Minor remarks:
- Introduction - add citation at the end of line 72…
- Results – add coma at lines 135-138
- I am confused… see Material, l. 233: “Solanum tuberosum L. subspecies andigena line 7540”, see Abstract - l. 20 – “Solanum tuberosum ssp. Andigena”, in Results, l. - line 93-94 “Solanum tuberosum ssp. tuberosum Désirée (data not shown) as well as from Solanum tuberosum ssp. Andigena (Fig. 1A, B)”, but 2 - “Solanum tuberosum variety Désirée” (l. 149), then what about Figs.3-4 – what variety/ subspecies is shown?
In addition, there are a mistakes in the text of manuscript, especially in Reference list (e.g. lines 321, 323, 326, 339, 345, 352, 356, 360) that need to be corrected by Authors.
Author Response
Point-by-point Response to Reviewer #2
Line 72: a reference is now given here.
Coma was added in lines 135-138 as suggested.
The reviewer is right and we provide now data for Solanum tuberosum spp. andigena (Fig. 1) and Solanum tuberosum ssp. tuberosum (Supplementary Fig. 2). The WT analyses shown in Figure 3 and 4 are done with Solanum tuberosum spp. tuberosum and this in now clearly stated.
All mistakes in the reference list have been corrected as suggested.
Reviewer 3 Report
Combining assays of wild Solanum tuberosum, and mutants over- and lower-expressing one sucrose transporter, the manuscript shows the sucrose-dependent increase of the level of miR172 involved as flower and tuber inducing mobile signal.
The display and readiness have been significantly improved in the new manuscript. Minor mistakes remain that I detail along with specific scientific questions.
- Confused use of brackets. Compare improper use in lines 38-39 with correct writing in lines 40-41.
- Figure 1. What RNAi 2/5, RNAi 2/16, OE8 and OE22 are? Is OE for overexpressing? Indicate them in the legend. In legend and text, "35S:" appears in several places without explanation of its meaning. I suppose it refers to the 35S RNA promotor used in the construction, and that must be indicated, at least in Materials and Methods.
- Caution must be added when interpreting the results. Plants overexpressing SUT4-GFP23 and SUT4-GFP14 are not the same than StSUT4-overexpressing plants. Does GFP label affect to sugar transport activity?
Author Response
Point-by-point Response to Reviewer #3
- The usage of brackets was corrected as suggested.
- The figure legend was now completed and it is now clearly stated in the legend as well as in the material and method section that the cauliflower mosaic virus CaMV35 S promoter was used for the generation of overexpressing plants (Martin et al. 2009).
- We now clearly stated that StSUT4-GFP overexpressing plants have been analyzed. Previous experiments showed us that a C-terminal fusion of GFP to sucrose transporters didn`t affect sucrose transport activity as revealed by yeast complementation experiments (Lalonde et al 2003). This is now mentioned in the text as well.
Round 2
Reviewer 1 Report
As mentioned in my last report, I still believe that this paper suffers from many flaws and the authors have not replied thoroughly to all my comments.
The biggest matter with this paper is the figure 1 which does not clearly shown a tight relarionship between the levels of StSUT4 and miRNA. Without this demonstartion, the core message of this paper will remain too much speculative.
Author Response
Response to reviewer #1:
Dear Reviewer #1,
Thank you very much to help us improving our manuscript.
As you suggested we corrected the introduction and added more experimental data from Solanum tuberosum andigena as well as from Solanum tuberosum tuberosum Désirée as you suggested, and supplied also results from the real time PCR analysis from shoot apical meristems as a additional supplement.
We added the data of glucose feeding experiments as additional supplement.
We performed statistical analysis of all data.
We rewrote the title of the manuscript and are now more careful in drawing our conclusions.
The only thing that is still missing is the real time qPCR of miR172 levels in sink leaves of miR172 overexpressing potato plants. The reason for this is that we need to perform these experiments (which are already published by Martin et al. 2009) which takes longer than 7 days.
Instead of repeating these analyses we provide new information about the diurnal pattern of miR172 accumulation that is increasing during the light period (supplementary figure 4) and significantly increased in one out of two transgenic StSUT4-silenced andigena plants showing high sugar accumulation at the end of the light period [2].
We are convinced to thereby support our main message which is the sugar-dependent regulation of miR172 in transgenic plants as well as in feeding experiments.
Round 3
Reviewer 1 Report
This new version of manuscript has highly been improved. The results section is now clearer and there is no major conflict/gap between the figures and interpretation.
Two points needs to be addressed:
1) As the figure 3 corresponds to the expression of miRNA172 in the whole in vitro plantlets treated with different sugars, I propose to put it together with those showing length and pictures of the in vitro plantlets. I mean a reorganization of the figure 3 and the figures 4A and 4B.
One possibility is to put according to the following order: Figure 3A) pictures of the whole in vitro plantlets, 3B) length measurements and 3C) miRNA expression
2) Statistical analysis is missing for supplemental results
Author Response
Point-by-point response to reviewer #1:
1) As the figure 3 corresponds to the expression of miRNA172 in the whole in vitro plantlets treated with different sugars, I propose to put it together with those showing length and pictures of the in vitro plantlets. I mean a reorganization of the figure 3 and the figures 4A and 4B.
One possibility is to put according to the following order: Figure 3A) pictures of the whole in vitro plantlets, 3B) length measurements and 3C) miRNA expression
We agree with reviewer #1 and rearranged both figures. The phenotypic characterization is now shown in Fig. 3 A and B as suggested, followed by the real time qPCR data of miR172 given in Fig. 3C. The text and figure legends have been changed accordingly.
2) Statistical analysis is missing for supplemental results
The reviewer is right. We performed statistical re-evaluation of all 4 figures supplied as "Supplementary data".
This manuscript is a resubmission of an earlier submission. The following is a list of the peer review reports and author responses from that submission.
Round 1
Reviewer 1 Report
Garg et al. investigated the regulation of the phloem-mobile miRNA 172, that is involved in both flowering and tuberization in potato, by sugars and thereby checked whether Arabidopsis and potato share the same regulatory mechanisms, regarding sugar-mediated the control of two miRNA : miRNA176 and miRNA156.
The unfolded experimental strategy involves the investigation of miRNA172 expression in two kind of mutants of the sugar transporter STSUT4 and sugar feeding experiments applied to whole plantlets and whole cut leaves. The choice of sugar transporter mutants matched well the physiological relevance of such a transporter in potato physiology.
In my opinion, this paper suffers many flaws that deserves to be addressed carefully.
1) The title does not match with the paper’s content/main message. In the title, the authors talk about the role of miRNA in sugar starvation, while neither the results about sugar status in different experimental conditions nor the analysis of the expression of the well-known markers of sugar starvation in different sugar-feeding experimentsand in mutants were conducted.
In addition, if the authors clearly want to tackle this topic, they have to describe previous works regarding miRNAs, sugar starvation and for instance SnRK1 regulation.
2) The introduction section is mainy focused on the central role of StSUT4 in potato physiology, at the expense of knowledge about miRNA, including miRNA172 and miRNA156. For example, how does sugar mediate the regultaion of these miRNA in Arabidopsis ? Is this regulation only limited to Arabidopsis or extended to other species ? Certain recent references are missing (Ponnu et al., 2020)
3) There is no statistical analysis to support the results depicted in all figures. In addition, the authors mentionned in Abstract (Lines 20 and 21), that glucose and fructose repress the expression of miRNA172. However, there is no results with fructose. Additionnal experiments with fructose alone, glucose + fructose are required to check whether the two sucrose-derivated hexoses are able or not to regulate miRNA172.
4) Figures 2A and 2B correspond to the same results and bring the same message. If not, please clarify the difference between them. In addition, one kinetic point (4h) is missing in the figre 2B for sorbitol. For the figure 3C, please provide for evey treatment, a picture depicting three biological repititions. How many plantlets were used for each treatment ?
5) The order of the figures was wrong. Figure 3 preceeds figure 2 !!!
6) Discussion is focused on the inability of sucrose to repress the miRNA156 expression, while such a result is only based on the one of the figure 4B. The relationship between sugar and miRNA172 was omitted. To support at least partially the hypothesis that miRNA156 expression is not repressed by sugar, additional experiments carried out in the SUT4 mutants and alos based on sugar feeding strategy are required.
Reviewer 2 Report
Manuscript entitled "The role of developmental miRNAs during sugar starvation in potato plants" intended for publication in International Journal of Molecular Sciences is generally an interesting short-communication paper. However, the manuscript is not suitable for publication in the present form and needs improvements.
I think that Authors should improve some parts of manuscript. Complete the affiliation address. In Material and methods Authors should add more information on plants growth conditions (e.g. light intensity and duration of growth), and add information on statistical analysis. Add also title to table presenting primers (it could be moved to supplementary material). The Authors should improve figures presentation. I mean, that especially figure captions must be described in a more detailed way, e.g. all abbreviations to Fig 1 must be explained and to Fig.3 statistical analysis should be added. The Authors must explain all abbreviation used in the manuscript, e.g. "LD conditions”… I think that Discussion and Conclusions could be modified. What do you mean using phrases: "seems to be repressed by", "was rather increased in response" or "The efflux of sucrose"? In addition, there are mistakes in the text of manuscript, especially in References list that need to be corrected by Authors (e.g. lines: 34, 115, 124, 212, 241, and 254). References must be carefully checked and improved.
Reviewer 3 Report
Several formal mistakes, ambiguous correspondence among the title, the text and the figures place the manuscript outside of the current scientific communication.
Authors must re-write and correct the display of the manuscript before an in deep evaluation.
I highlight only a few of the many aspects that must be addressed and corrected.
Figures. They must be displayed in sequential order and properly referred in the text (among others, see the jumble of figure 2). What colours mean in Figure 3? Quality colour is poor in several figures. Statistical treatment of data must be indicated in legends and explained in Materials and Methods.
English must be thoroughly revised. As examples: text in lines 35-38 (check brackets!).
Objectives are poorly described in paragraph of lines 59-66. New terms as, transcription factors, are introduced without references. The manuscript fails to address the interest of the objectives in the face of the present knowledge in the field.
Materials and Methods. In addition of the statistical analyses, methods used of RNA extraction and quantification must be described.